# Antibody Response to ChAdOx1 nCoV-19 (AZD1222) Vaccine in Kidney Transplant Recipients

**DOI:** 10.3390/vaccines10101693

**Published:** 2022-10-11

**Authors:** Dharmendra S. Bhadauria, Harshita Katiyar, Amit Goel, Prachi Tiwari, Ravi V. Krishna Kishore, Amita Aggarwal, Alka Verma, Dheeraj Khetan, Anupma Kaul, Monika Yachha, Manas Ranjan Behera, Brijesh Yadav, Narayan Prasad

**Affiliations:** 1Department of Nephrology, Sanjay Gandhi Postgraduate Institute of Medical Sciences, Lucknow 226014, India; 2Gastroenterology, Sanjay Gandhi Postgraduate Institute of Medical Sciences, Lucknow 226014, India; 3Clinical Immunology, Sanjay Gandhi Postgraduate Institute of Medical Sciences, Lucknow 226014, India; 4Department of Emergency Medicine, Sanjay Gandhi Postgraduate Institute of Medical Sciences, Lucknow 226014, India; 5Department of Transfusion medicine, Sanjay Gandhi Postgraduate Institute of Medical Sciences, Lucknow 226014, India

**Keywords:** vaccination, anti-SARS-CoV-2 antibody, humoral immunity, Covishield, neutralising antibodies, kidney transplant recipients

## Abstract

Kidney transplant recipients (KTRs) are at a much higher risk of complications and death following COVID-19 and are poor vaccine responders. The data are limited on the immune response to Covishield^®^ in KTRs. We prospectively recruited a cohort of 67 KTRs aged >18 between April 2021 and December 2021. Each participant was given two intramuscular doses of Covishield^®^, each of 0.5 mL, at an interval of 12 weeks. A blood specimen of 5.0 mL was collected from each participant at two points within a few days before administering the first dose of the vaccine and at any time between 4–12 weeks after administering the second dose. The sera were tested for anti-RBD antibody (ARAb) titre and neutralising antibody (NAb). An ACE2 competition assay was used as a proxy for virus neutralization. According to the prior COVID-19 infection, participants were grouped as (i) group A: prior symptomatic COVID-19 infection, (ii) group B: prior asymptomatic COVID-19 infection as evidenced by detectable ARAb in the prevaccination specimen, (iii) Group C: no prior infection with COVID-19, (iv) group D: Unclassified, i.e., participants had no symptoms suggestive of COVID-19, but their prevaccination specimen was not available for ARAb testing before vaccination. Fifty of sixty-seven participants (74.6%) provided paired specimens (group A 14, group B 27, and group C 9) and 17 participants (25.4%) provided only postvaccination specimens (group D). In the overall cohort (*n* = 67), 91% and 77.6% of participants developed ARAb and NAb, respectively. Their ARAb titre and NAb proportion were 2927 (520–7124) U/mL and 87.9 (24.4–93.2) %, respectively. Their median ARAb titre increased 65.6 folds, from 38.2 U/mL to 3137 U/mL. Similarly, the proportion of participants with NAb increased from 56% to 86%, and the NAb proportion raised 2.7 folds, from 23% to 91%. A comparison of vaccine response between the study groups showed that all those with or without prior COVID-19 infection showed a significant rise in ARAb titre (*p* < 0.05) and NAb proportion (*p* < 0.05) after the two doses of vaccine administration. The median value of folds rise in anti-RBD and NAb between groups A and B were comparable. Hence, ARAb is present in more than 3/4th of KTRs before the ChAdOx1 vaccine in India. The titer of ARAb and the proportion of NAb significantly increased after the two doses of the ChAdOx1 vaccine in KTRs.

## 1. Introduction

The COVID-19 pandemic has cost millions of lives globally. The scientific community and industry worked together to combat the situation and developed various vaccines, using different strategies against the novel coronavirus [1]. These vaccines have saved enormous spending and millions of lives [2]. The ChAdOx1 nCoV-19 vaccine (AZD1222), based on a replication-deficient chimpanzee adenoviral vector ChAdOx1, contains the SARS-CoV-2 structural surface glycoprotein antigen (spike protein) gene, in one among the various vaccines, which have shown promising protective efficacy in large human trials [3,4]. This vaccine, available under the brand name Covishield^®^, has been widely used in India. Over 1500 million doses of Covishield^®^ have been administered in India alone.

COVID-19 had a high risk of death for people with comorbidities, such as diabetes mellitus, obesity, hypertension, coronary artery disease, advanced renal failure, and the immunosuppressed population [5]. People with kidney diseases, such as chronic kidney disease, on maintenance haemodialysis, or KTRs, are at a much higher risk of complications and death following COVID-19 [6,7]. Considering the enhanced risk of organ dysfunction and mortality, it is strongly recommended to prioritise the COVID-19 vaccination of KTRs patients [8,9].

Unfortunately, the data are limited on the immune response to Covishield^®^ in KTRs. We studied the serological immune response after administering a complete two-dose schedule of the Covishield^®^ vaccine in KTRs.

## 2. Methods

This prospective observational cohort study was conducted between June 2021 and December 2021 in the department of Nephrology. The adults (age >18) who had undergone renal transplantation at least 12 months prior and had stable renal function were counselled to receive the anti-Covid vaccine. All KTRs were maintained on triple-drug immunosuppression consisting of Tacrolimus/cyclosporine (CNI), Mycophenolate mofetil (MMF), and prednisolone, as suggested in KDIGO guidelines [10].

Patients were included in the study if they had either not had even a single dose of a vaccine or had received two doses of the Covishield^®^ vaccine. People were excluded if they had either, (i) covid infection, confirmed with the nucleic acid test in the last eight weeks, (ii) received two doses of the Covishield^®^ vaccine at an interval of more than six months, (iii) had received another COVID-19 vaccine, in addition to Covishield^®^, or (iv) if six or more months had passed since the administration of the second dose of Covishield^®^.

Relevant data were collected on a predesigned data collection form. Each participant was given two intramuscular doses of Covishield^®^, each of 0.5 mL, at an interval of 12 weeks. The vaccine administration was carried out in the vaccination facility established at our institute by the state government to vaccinate common citizens. The vaccine was administered according to the standard precautions laid out by the state government, and the participants were observed on-site for 30 min after each dose. All participants were in telephone contact for the next 48 h to report any significant adverse events after vaccination.

Specimen collection: A blood specimen of 5.0 mL was collected from each unvaccinated participant at two time points within a few days before administering the first dose of the vaccine and at any time between 4–12 weeks after the administration of the second dose. Only a second blood specimen was collected from those who were completely vaccinated. The serum was separated by centrifugation at 4000× *g* for 10 min at 4 °C within 1 h of blood collection and was stored in multiple aliquots in deep freezers at −80 °C temperature. These stored sera were used for serological testing at the end of the study.

Study groups: According to the prior COVID-19 infection, participants were grouped as (i) group A: prior symptomatic COVID-19 infection, i.e., participants had nucleic acid test-confirmed symptomatic COVID-19 infection before vaccination, (ii) group B: prior asymptomatic COVID-19 infection as evidenced by detectable ARAb in a prevaccination specimen (iii) group C: no prior infection with COVID-19, i.e., their prevaccination specimen tested negative for ARAb, and (iv) group D: Unclassified, i.e., participants had no symptoms suggestive of COVID-19, but their prevaccination specimen was not available for ARAb testing before vaccination.

Serological testing: Stored sera were tested for anti-RBD antibody (ARAb) titre and neutralising antibody (NAb) using Elecsys Anti-SARS-CoV-2 S (Roche Diagnostics GmbH, Sandhofer Strasse 116, D-68305 Mannheim) and a SARS-CoV-2 Neutralizing Antibody ELISA Kit (Invitrogen, Thermo-Fischer, Catalog no BMS2326), respectively. Elecsys Anti-SARS-CoV-2 S immunoassay is used for in vitro quantitative assay for the antibodies against spike (S) protein receptor-binding domain (RBD) in humans. The test principle is based on an automated system’s double-antigen sandwich assay format. The assay had a sensitivity of 98.8% and a specificity of 99.98%. The limit of quantitation for the assay is 0.40–250 U/mL. The specimens with an antibody titre above 250 U/mL were serially diluted 20, 50, and 100 folds to get the results within the detection range. Titres which remained above the quantitation limit after 100-fold dilutions were reported as >25,000 U/mL. The antibody concentrations are expressed as U/mL, and a value ≥ 0.80 U/mL is considered positive for the anti-SARS-CoV-2 anti-RBD antibody. The SARS-CoV-2 Neutralizing Antibody ELISA Kit is an ACE2 competitive ELISA assay which was used as a proxy for virus neutralization. The serum was diluted to a 1:50 ratio with assay buffer as recommended by the Kit protocol. The specimens with a calculated neutralisation ≥ 20% were considered positive. Both assays were performed following the manufacturer’s recommendations.

Qualitative and quantitative data are expressed as numbers, proportions, and median (interquartile range). Paired numerical data are compared using a nonparametric Wilcoxon signed-rank test. The data between the groups were compared using the Mann–Whitney test. The analyses were performed using STATA 16 software. The level of significance was kept at <0.05.

The institute ethics committee approved the study, and the participants were enrolled after obtaining written informed consent.

## 3. Results

The study included sixty-seven participants (males 93%; age 37 (31–48) years). The clinical and laboratory details of the participants are summarised in Table 1. The participants had undergone organ transplantation 51 (34–74) months prior and had received two doses of the vaccine at an interval of 13 (12–14) weeks, and follow-up blood specimens were collected after 8 (5–11) weeks after the administration of the second doses of Covishield^®^.

Fifty participants (74.6%) provided paired specimens (group A 14, group B 27, and group C 9) and 17 participants (25.4%) provided only postvaccination specimens (group D). All 117 sera were tested for ARAb and NAb assay. In the overall cohort (*n* = 67), 91% and 77.6% of participants developed ARAb and NAb, respectively. Their ARAb titre and NAb proportion were 2927 (520–7124) U/mL and 87.9 (24.4–93.2)%, respectively.

Of the fifty participants who provided their paired serum specimens, three in group C and one in group A failed to develop ARAb after vaccination. The proportion of participants who tested positive for ARAb marginally increased from 78% to 92%. Their median ARAb titre increased 65.6 folds, from 38.2 U/mL to 3137 U/mL. Similarly, the proportion of participants with NAb increased from 56% to 86%, and the NAb proportion raised 2.7 folds, from 23% to 91% (Table 2).

A comparison of vaccine response between the study groups showed that all those with or without prior covid infection showed a significant rise in ARAb titre (*p* < 0.05) and NAb proportion (*p* < 0.05) after the two doses of vaccine administration (Table 3).

The median value of fold rise in anti-RBD and neutralising antibodies between group A (100.3 folds and 1.7 folds, respectively), and B (37.6 folds and 2.8 folds, respectively), were comparable, as shown in Figure 1A–D. The titre, anti-RBD and NAb positively correlated for overall samples (spearman r = 0.66, *p* < 0.001). Further, group-wise, the postvaccination titre of anti-RBD and NAb were positively correlated. For Gp A (r = 0.684, *p* = 0.007), Gp B (r= 0.741, *p* < 0.001), Gp C (r = 0.745, *p* = 0.021), Gp D (r = 0.292, *p* = 0.256).

## 4. Discussion

We studied the serological immune response to a two-dose schedule of the ChAdOx1 nCoV-19 vaccine (AZD1222) in 67 kidney transplant recipients in a real-life setting. Following two vaccine doses, 91% and 77.6% of participants developed ARAb and NAb, respectively. Their ARAb titre and NAb proportion were 2927 (520–7124) U/mL and 87.9 (24.4–93.2) %, respectively.

Since the introduction of vaccines, a large number of data have emerged on the immune response to the COVID-19 vaccine among the low-risk healthy population and high-risk immunocompromised population, including KTRs. A recent systematic review of 112 studies on immune response among solid organ transplant recipients (SOTR) showed that immune response, compared to healthy controls, is relatively impaired in SOTR. Most of the included studies (88/112) included KTRs. Only 44% of SOTR showed an immune response following vaccination. The response was impaired, regardless of the type of vaccine used or the nature of organ transplantation [11]. The immune response was relatively better in liver transplant recipients (68%) compared to kidney (40%) and heart (45%) transplant recipients. Further, the response rate for ChAdOx1 nCoV-19 vaccine was lower (36%) compared to that with the mRNA-12723 mRNA 52% and BNT162b2 vaccine (43%) [11].

We have limited data on the immune response to the ChAdOx1 nCoV-19 vaccine (AZD1222) in the KTRs population. Our study has shown that a large proportion of KTRs had an asymptomatic infection before vaccination. The balance of participants with asymptomatic infection was 78% in our study, which is comparable to 90% found by Prasad et al. in another Indian study [12]. A large study has shown that 90% of infections in India are asymptomatic [13]. An excellent systematic review of global data suggested that 35% of the disease remains asymptomatic [14]. This significant difference in the proportion of asymptomatic infection may be related to the viral strain, host immunogenicity, associated comorbidities, ethnicity or several other unexplored factors.

We found that 67% of those without prior infection seroconverted after the two doses of vaccine. Similarly, a seroconversion rate of 44% was observed after the ChAdOx1 vaccination in UK-based KTRs [15], and a lower IgG titre was observed in the German solid organ transplant cohort [16]. Kute et al. reported 61.2% seroconversion in uninfected KTRs in an Indian cohort [17]. Similarly, Jasuja et al. from India reported around 49% seroconversion in a similar cohort of uninfected KTRs [18]. This difference could be related to the variation in the study population or the diagnostic assay used to detect ARAb and neutralising antibodies in these studies.

Compared to other studies, which studied the immune response to the ChAdOx1 vaccine in KTRs, we analysed the ARAb titres and NAb in paired specimens. The paired data analysis helped us to understand the vaccine’s booster effect on ARAb titre and its neutralising capacity. The median rise in titre of ARAb and percentage of NAb after the second vaccination dose was around 36 and 3-fold, respectively. A significant increase in anti-RBD and neutralising antibodies was observed after complete vaccination, regardless of prior COVID-19 infection. Previous studies have demonstrated a similar pattern of SARS-CoV-2 antibodies in KTRs after natural COVID-19 [19,20,21]. Natural infection with SARS-CoV-2 usually triggers the generation of high titres of protective neutralising anti-spike IgGs in KTRs [22]. The KTRs with detectable antibodies in the prevaccination specimen experienced a robust postvaccination immunological response due to past COVID-19 infection-related immune memory persisting for a prolonged period [23,24,25] and a quick, complete response to COVID-19 vaccines [26]. The protective role of these antibodies is not completely understood. One study carried out among healthcare workers suggested a threshold titre of anti-RBD and neutralising antibodies that were ≥142 and > 62 BAU/mL, respectively, to ensure >90% protection against symptomatic COVID-19 [27].

None of the studies has assessed the response of NAb in paired specimens following the ChAdOx1 vaccine in KTRs. In our study, neutralising antibodies were present in about half of KTRs having a prior infection, which increased to over 90% after vaccination. Whether anti-RBD antibodies can protect against COVID-19 is not completely known. However, neutralising antibodies can determine whether a booster vaccine is required or if seropositive KTRs should have pre-exposure neutralising monoclonal antibodies [27].

Our study had strengths in paired specimen testing and the determination of neutralising antibodies in paired serum specimens. This study has a few limitations: first, a small number of participants; a lack of serological response measurement after the first dose of the vaccine; third, the noninclusion of a control population and the neutralising antibodies that we measured were semiquantitative.

## 5. Conclusions

ARAb is present in more than 3/4th of KTRs before the ChAdOx1 vaccine in India. The titre of ARAb and the proportion of NAb significantly increased after the two doses of the ChAdOx1 vaccine in KTRs.

## Figures and Tables

**Figure 1 vaccines-10-01693-f001:**
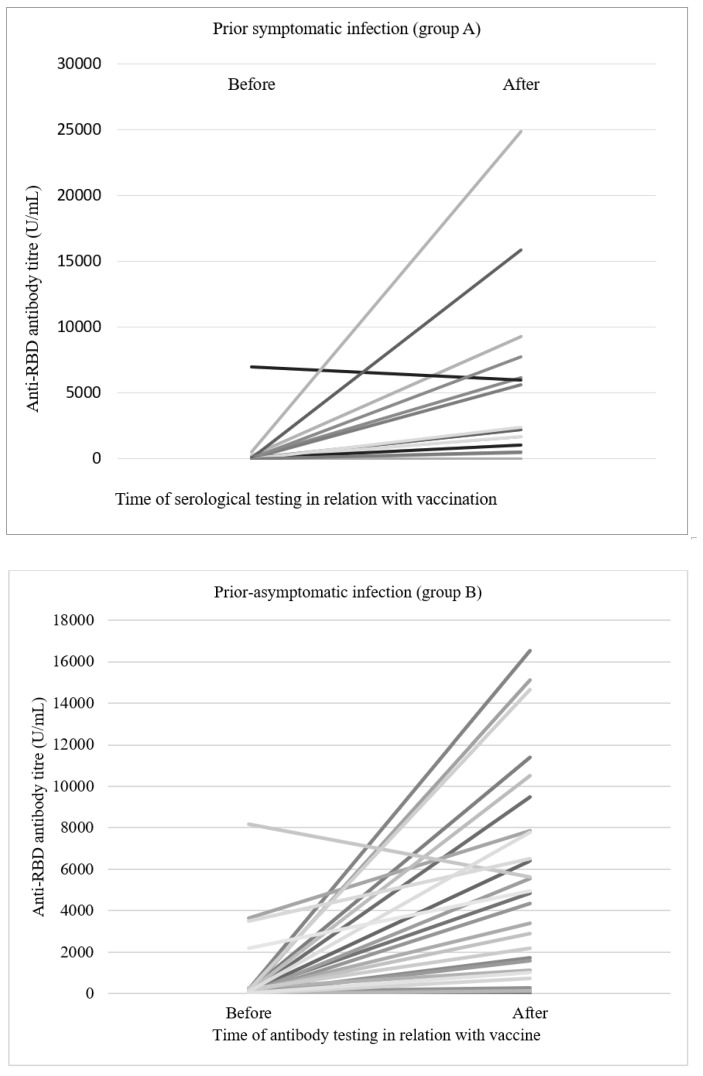
Figure 1(**A**–**C**). Anti-RBD antibodies titer of individual participants, before and after vaccine in different subgroups of renal allograft recipients. 1(**D**). Anti RBD antibodies titer in groups of renal transplant recipients with symptomatic or asymptomatic prior Covid infection.

**Table 1 vaccines-10-01693-t001:** Characteristics of the study participants (*n* = 67).

Variable	Values
Males	62 (93)
Age (years)	38 (31–48)
Haemoglobin (g/dL)	13.6 (12.3–15.0)
While cell counts (×1000 mm^3^)	7.6 (6.0–10.3)
Platelets counts (×1000 µL)	183 (137–231)
Serum creatinine (mg/dL)	1.3 (1.0–1.6)
Serum total protein (g/dL)	6.6 (6.2–7.2)
Serum albumin (g/dL)	4.2 (3.9–4.6)
Interval since transplantation (months)	51 (34–74)
Maintenance immunosuppression Tacrolimus/mycophenolate/Prednisolone Cyclosporin A/Mycophenplate/Prednisolone	59 (88) 08 (12)
The interval between two doses (weeks)	13 (12–14)
The interval between the second dose and specimen collection (weeks)	8 (5–11)

Categorical data are presented as numbers and proportions; numerical data are expressed as median (interquartile range).

**Table 2 vaccines-10-01693-t002:** COVID-19 anti-RBD and neutralising antibody before and after two doses of vaccination (*n* = 50 pairs).

Type of Antibody	Qualitative Results		Quantitative Results		
	Before Vaccination	After Vaccination	Before Vaccination	After Vaccination	Folds Elevation
Anti-RBD antibody	39 (78)	46 (92)	38.2 (2.4–122.7)	3137 (638.3–7748.0)	65.6 (8.8–248.3)
Neutralising antibody	28 (56)	43 (86)	23.0 (11.3–52.0)	91.3 (63.2–93.3)	2.7 (1.3–4.7)

The qualitative and quantitative data are expressed as number (proportion) and median (interquartile range). The titre of the anti-RBD antibody is expressed as U/mL, and neutralising antibody is expressed as %. Qualitative seroconversion was determined based on SARS-CoV-2 anti-RBD antibody titre ≥ 0.80 U/mL.

**Table 3 vaccines-10-01693-t003:** Comparison of COVID-19 anti-RBD and neutralising antibody before and after two doses of vaccination among different study groups.

Type of Antibody	Prior Symptomatic Infection (Group A, *n* = 14)		*p*-Value	Prior Asymptomatic Infection (Group B, *n* = 27)		*p*-Value	No Prior Infection (Group C, *n* = 9)		*p*-Value
	Before Vaccine	After Vaccine		Before Vaccine	After Vaccine		Before Vaccine	After Vaccine	
Anti-RBD antibody	36.2 (2.4–88.7)	3995 (1056–7748)	<0.05	104.8 (14.0–208.1)	4840 (1121–7874)	<0.05	0.4 (0.4–0.4)	29.2 (0.6–1143)	<0.05
Neutralising antibody	40.5 (12.7–71.2)	92.0 (75.6–93.2)	<0.05	24.8 (19.8–52.0)	92.0 (78.6–93.9)	<0.05	7.2 (6.3–11.3)	26.1 (14.4–67.0)	<0.05

The titre of the anti-RBD antibody is expressed as U/mL, and neutralising antibody is expressed as %. The qualitative and quantitative data are expressed as number (proportion) and median (interquartile range).

## Data Availability

The data presented in this study are available with the corresponding author of the manuscript and the same can be reached on reasonable request. The data are not publicly available due to ethical issue.

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
