# Peer review of "Antibody Response to ChAdOx1 nCoV-19 (AZD1222) Vaccine in Kidney Transplant Recipients"

_vaccines, 2022, doi:10.3390/vaccines10101693_

Round 1

Reviewer 1 Report

The manuscript submitted by Bhadauria et al. investigates the immune response upon ChAdOx1 nCov-19 vaccination in kidney transplant recipients. This is a population at high risk for severe Covid and known to have inadequate vaccine responses. As such the research performed by Bhadauria et al. is highly relevant. The strength of the manuscript is the use of paired pre- and post-vaccination samples.  I do have some remarks that should be addressed to make the manuscript appropriate for publication.

1.     My major concern is that a more in depth analysis should be performed on these valuable samples. From my understanding the neutralization assay is performed using a single serum concentration (so semi-quantitative) whereas the anti-Spike antibody responses were analyzed using dilution series of the sera (at least of those containing high antibody titers). As levels of neutralizing serum antibodies is for now the best correlate for protection, it is important that these levels are quantified. So for the serum samples that display more than 50% calculated neutralization additional serum dilutions should be tested until no neutralization can be observed. This allows the determination of a IC50 value for the sera with considerable neutralization activity. Having these data, the % of sera that have more than 20% and more than 50% calculated neutralization activity at the highest tested serum concentration can be reported, next to the IC50 of the sera with (at least for those sera with high levels of neutralizing antibodies). As such informative correlations can be tested, including:

-        What is the correlation between anti-spike antibody levels and neutralization activity for all samples?

-        What is the correlation between anti-spike antibody levels and neutralization activity per group post vaccination? Are there clear differences between groups?

-        Is there a difference in correlation between sera of persons that have only been vaccinated and not infected (post-vaccination sera of group C) and sera of persons that have been infected but not yet vaccinated (pre-vaccination group A and B)?

-        For the quantitative analysis please provide graphs containing spike specific antibody titers for each individual sample. Preferably these can be shown as linked data-point for pre- and post-vaccination.

2.     The study lacks groups of healthy persons. Therefore, it is not clear to which extend the immune responses of the kidney transplant patients are blunted. To have at least a estimate it is highly advised to include immune serum of healthy vaccinees or convalescent sera of healthy persons that were previously infected as positive control and reference.

3.     In the neutralization assay, which serum dilution was used ? Please, indicate this in the material and methods.

4.     Please clarify why in the neutralization assay 20% neutralization is used as limit to consider a sample positive for neutralization activity.

5.     Please indicated in the abstract and result section that an ACE2 competition assay was used as a proxy for virus neutralization. This is important as also antibodies that cannot compete with ACE2 for RBD binding can be neutralizing

6.     The sandwhich ELISA used to detect Spike specific serum antibodies is based on the RBD domain of the spike. Therefore, in the text anti-RBD antibody instead of anti-Spike antibody should be used.

In discussion meta analysis overall 40% response vaccines but only 0.39% for ChAdOX1. However in this study much higher reponses (qualitative) please clarify: diff in qualitative limits?

Author Response

Thanks for your critical review and giving us the opportunity to improve the manuscript.

Kindly find the attachment of your comment reply.

Reviewer 2 Report

Authors wrote that blood samples were collected from each participant at two-time points, but in the same time patients who were included in group D did not have a pre-vaccination specimen.

 Please:

1         rephrase the text

2         Determine the inclusion and exclusion criteria for each group.

Regarding the methods of the study, the time span for the 2nd blood specimen (is very big 4-12 wks).

I would like to propose to assess the results regarding three different time points (for example: 4 wks, > 4 wks and < 8 wks, and > 8 to 12 wks).  Other wiles the results not valid.

Table 2 is not comprehensive

Pleas e define the terms “Qualitative and quantitative”, because the Roche reagents are only for quantitative detection of antibodies.  

Do you use any other kit for qualitative test?

Table 3. Please correct “Prio”

From table 3 I presume that asymptomatic patients had higher titer, before and after vaccination.  Please present the results of group A in comparison to group B as graphic.

Please write COVID-19 in the same manner in the text. You use COVID-19, covid 19, and Covid 19 in the text.

Conclusion section is missing in this article. 

Author Response

Thanks for your critical review to improve our manuscript. 

Find herewith the attachment. Hoping that we have addressed your concerns in the revised manuscript satisfactorily. 

Round 2

Reviewer 1 Report

Dear authors

Thanks for your response to my comments.

I understand that you do not have the means do test multiple dilutions of the sera. As such I would highly recommend to indicate in the discussion that the neutralization assay is semi-quantitative.

I noticed in the pdf "with the authors response" that comment 1d lacks a response. Possible this is due to a pdf conversion issue.

For comment 1.e the adapted figure 1.A-C is not included int the adapted manuscript. Please provide the figure to me

Author Response

Kindly find herewith our response to your comments and revised version of the manuscript.

Reviewer 2 Report

All my previous comments and suggestions were answered satisfactory by authors.

Author Response

Thank you for accepting our reply to your comments and accepting manuscript.
